# QUADRo: Dataset and Models for QUestion-Answer Database Retrieval

**Stefano Campese**[*]
University of Trento
Amazon Alexa AI
stefano.campese@unitn.it

**Ivano Lauriola**
Amazon Alexa AI
lauivano@amazon.com

**Alessandro Moschitti**
Amazon Alexa AI
amosch@amazon.com

## Abstract

An effective approach to design automated Question Answering (QA) systems is to efficiently retrieve answers from pre-computed databases containing question/answer pairs. One of the main challenges to this design is the lack of training/testing data. Existing resources are limited in size and topics and either do not consider answers (question-question similarity only) or their quality in the annotation process. To fill this gap, we introduce a novel open-domain annotated resource to train and evaluate models for this task. The resource consists of 15,211 input questions. Each question is paired with 30 similar question/answer pairs, resulting in a total of 443,000 annotated examples. The binary label associated with each pair indicates the relevance with respect to the input question. Furthermore, we report extensive experimentation to test the quality and properties of our resource with respect to various key aspects of QA systems, including answer relevance, training strategies, and models input configuration.

## 1 Introduction

Recently, two main QA paradigms have gained more and more attention from both the industrial and research community: open- and closed-book. The former paradigm, also known as *retrieve-and-read* refers to a set of models and techniques that, given an input question, follow a two-step process. First, they *retrieve* relevant content from a large corpus of documents, e.g. from the web. Second, they extract the answer from the retrieved content ([Yang et al., 2015](); [Nguyen et al., 2016](); [Thorne et al., 2018]()). The latter approach, closed-book, is relatively new and it consists of models that rely on knowledge acquired during training, and they generate an answer without accessing external corpora. Typically, these techniques are based on Seq2Seq models such as T5 ([Raffel et al., 2020]()). Although

these approaches have recently shown impressive performance, their execution places a substantial strain on system resources, making their adoption unpractical for industrial applications ([Chen et al., 2023](); [Wang et al., 2023]()). A third approach that started to become popular in the last years consists in retrieving the answer from a DataBase of pre-generated question/answer pairs rather than leveraging knowledge stored in the models' parameters or large corpora. These systems typically employ 3 components: (i) a DataBase of questions and their answers, (ii) a retrieval model to query the DataBase and to retrieve relevant content for an input question, and (iii) a ranking model to select the final answer. Recent literature showed various benefits of these approaches, here named DataBase QA (DBQA), including fast inference and the ability to introduce new knowledge by simply adding new pairs without re-training models.

One of the main issues with DBQA approaches is the lack of large training data to support the development of retrieval and ranking models. Only limited resources exist in the literature and often models are indeed trained on general tasks, including paraphrasing or semantic similarity ([Lewis et al., 2021]()). To fill this gap, we introduce a novel large annotated dataset to train and evaluate models for question ranking and DBQA. Our resource consists of 15,211 open-domain input questions. For each input question, we annotated 30 relevant q/a pairs and we marked them as semantically equivalent or not (binary label) with respect to the input. In total, the dataset comprises $\approx 443,000$ annotated examples. We selected q/a pairs using a DBQA model trained on publicly available data. This allowed us to easily integrate labeled hard negatives, making the resource significantly complex and challenging. Differently from existing resources mainly based on question-question pairs, our dataset is annotated both with respect to question-question similarity and question-answer correctness. To the best of

---

[*]Work done as an intern at Amazon Alexa AI

our knowledge, this resource is the largest annotated dataset for training DBQA models. We performed extensive experiments, which provide baselines and a comprehensive analysis of distinctive aspects, weaknesses, and benefits of the dataset.

As second contribution of this paper, we use the annotated resource to train models for DBQA. Then, we build a standard DBQA pipeline, named QUADRo (QUestion Answer Database Retrieval) based on these models and we test it in various open domain QA settings. Our experiments aim to address a critical gap in the existing literature by providing consolidated evidence regarding some key aspects of DBQA end-to-end pipelines. For instance, despite the existing evidence supporting the contribution of the answer to the retrieval task (Wang et al., 2020), its utilization remains poorly explored in DBQA applications (Seonwoo et al., 2022; Chowdhary et al., 2023).

Our annotated dataset and our models will be available to the research community[1].

## 2 Related Work

**Question Similarity and Ranking** Duplicate Question Detection (DQD) is a well known problem under the umbrella of Semantic-Textual-Similarity (STS) tasks. It aims at identifying when two questions are semantically equivalent or not. Early approaches focused on the extraction and creation of several types of lexical (Cai et al., 2011), syntactic (Moschitti, 2006) and heuristic (Filice et al., 2017) features to measure the similarity between two questions. Lately, translation- and topic-based modeling approaches have been explored, e.g., (Wu and Lan, 2016). DQD received a huge boost with the advent of embedding representations such as Word2Vec, Glove (Charlet and Damnati, 2017), and ELMO (Peters et al., 2018) to compute sentence-level embeddings for the two individual questions (Fadel et al., 2019).

More recently, pretrained Transformers set the state of the art for STS and DQD (Chandrasekaran and Mago, 2021). Peinelt et al. (2020) proposed tBERT, a Transformer model that takes the concatenation of the two questions as input, providing a joint and contextualized representation of the pair.

**DataBase QA** is a known paradigm that typically relies on (i) a curated database (or collection) of questions and their answers, (ii) a retrieval model to query the database (e.g. BM25 or DPR) for finding an equivalent question, and (iii) an optional ranking component.

Early works on DBQA for forum or FAQ (Nakov et al., 2016; Shen et al., 2017; Hoogeveen et al., 2015) have pointed out that when answers are available in the DataBase together with the questions, the resulting system can be very accurate. The main challenge to their general usage is the typical specificity of the DB, also associated with a limited availability of q/a pairs. Othman et al. (2019) introduced WEKOS, a system able to identify semantically equivalent questions from a FAQ database. The model, based on k-means clustering, Word-Embeddings (CBoW), and heuristics, was tested on a dataset based on Yahoo Answer, showing impressive performance compared to the existing systems. After the rise of Transformer models, Mass et al. (2020) proposed a new ensemble system that combines BM25 with BERT (Devlin et al., 2019) for retrieval and ranking. The same authors also explored systems based on a GPT2 model to generate question and/or answer paraphrases to fine-tune the BERT models on low-resource FAQ datasets with promising results. Similarly, Sakata et al. (2019) proposed a method based on BERT and TSUBAKI, an efficient retrieval architecture based on BM25 (Shinzato et al., 2012), to retrieve from a database of FAQ the most similar questions looking at query-question similarity. However, most of these works and practical applications were confined to specific domains, including domain-specific FAQ or Community QA.

Recently, effective open-domain DBQA pipelines have been shown. Lewis et al. (2021) assembled a DataBase of 65M q/a pairs automatically generated from the web. Then, they employed a neural retrieval and ranker based on Transformer models to query the DataBase. The authors compared this end-to-end DBQA pipeline, named RePAQ, against various closed- and open-book QA strategies, showing interesting performance in terms of efficiency and competitive accuracy. However, there are some limitations of that system as (i) models are not trained on annotated data for question ranking due to lack of available resources and (ii) the DataBase is noisy (estimated 18% incorrect q/a pairs). As an extension of this approach, Seonwoo et al. (2022) proposed a 2 steps retrieval based on BM25 and DPR executed in sequence to improve the

---

[1] https://github.com/amazon-science/question-ranking

efficiency. However, most of the technical details are missing, including the DataBase used and training mechanisms. It is worth noticing that these end-to-end systems rely solely on models trained for question similarity, without providing empirical evidence of the advantages gained from incorporating answers to enhance accuracy.

**Datasets and resources** Various resources have been made available to train models for DQD and DBQA. One of the most popular resources is the QuoraQP dataset. This consists of 404,290 pairs of questions extracted from the Quora website, and annotated as having the same meaning or not. An extension of the original dataset was released by Wang et al. (2020), which consists of the original question pairs concatenated with answers to the second questions, extracted from the original Quora threads. Another popular resource is the CQADup-Stack (Hoogeveen et al., 2015) dataset, originally released as a benchmark for Multi-Domain Community QA. It consists of questions coming from different threads sampled from twelve StackExchange subforums. The dataset contains annotations for similar and related questions. For a small portion of question pairs, there is an annotated answer. On average the dataset contains $\approx 5.03\%$ of duplicated questions. Next, WikiAnswers was built by clustering together questions classified as paraphrases by WikiAnswers users. The dataset consists of 30,370,994 clusters containing an average of 25 questions per cluster. Unfortunately, the answers are large paragraphs, which are not necessarily suitable for DQD. SemEval-2016 Task 3 challenge (Nakov et al., 2016) introduced another famous resource for DQD in community QA, where Question-Comment, Question-External Comment, and Question-Question Similarity, are annotated. Although this dataset can be used in a reranking setting, the amount of queries is limited. Moreover, all the data is extracted from specific domains.

In general, the resources above have some limitations: (i) the majority of datasets do not include answers, (ii) there are no guarantees on the quality, and (iii) with the exception of SemEval, these resources are based on pairs of questions (e.g., QuoraQP) rather than question ranking, preventing the possibility to study search and ranking problems. Our annotated dataset instead enables research on large-scale retrieval of semantically equivalent questions, associated with correct answers, which are essential to build large-scale DBQA models.

## 3 DataBase QA architecture

Inspired by previous work (Seonwoo et al., 2022; Lewis et al., 2021), we consider a retrieval-reranking DBQA architecture consisting of a large-scale database of questions and correct answers, an efficient search engine to query the database, and an answer selector (ranker). In the remainder of the paper, we call this DBQA system QUADRo (QUestion Answer Database Retrieval).

**Search Engine** This is based on recent findings in neural retrieval, including Dense Passage Retrieval (Karpukhin et al., 2020). It consists of a siamese Bi-Encoder Transformer (Reimers and Gurevych, 2019) (also known as Sentence-Transformer) network. The first branch encodes the input question $t$ as *[CLS] t [EOS]*, whereas the second branch encodes question/answer pairs $(q_i, a_i)$ from the database as *[CLS] $q_i$ [SEP] $a_i$ [EOS]*. The cosine similarity between the representations extracted from the two branches expresses the level of semantic similarity between the question and the q/a pair. Let $\delta : \Sigma* \rightarrow \mathbb{R}^d$ be the Transformer function which maps a text $s \in \Sigma*$ (either a question $q_i \in \mathcal{Q}$, an answer $a_i \in \mathcal{A}$, or a pair $(q_i, a_i) \in \mathcal{Q} \times \mathcal{A}$) into a $d$-dimensional embedding. The final score assigned to a target question $t \in \mathcal{Q}$ and an element of the database $(q_i, a_i)$ is $\frac{\delta(t)^\top \delta(q_i, a_i)}{\|\delta(t)\|\|\delta(q_i, a_i)\|}$, where $\|\cdot\|$ denotes the 2-norm of a vector. When the user asks a new question, the bi-encoder computes the cosine distance between the question embedding and all q/a pairs and returns the most similar $k$ pairs.

**Answer Selector** After the retrieval stage, an answer selector (or reranker) model re-ranks the pairs returned by the search engine and selects the final answer to be served to the user. Formally, let $\mathcal{R} = \{(q_i, a_i)\}_{i=1}^k$ be the set of $k$ returned pairs for a given target question $t$. The answer selector $r : \mathcal{Q} \times \mathcal{R} \rightarrow \mathbb{R}$ assigns a score to each triplet, $r(t, q_i, a_i)$, and returns the answer associated with highest ranked pair, i.e., $\arg\max_{i=1...k} r(t, q_i, a_i)$. Inspired by the Contextual Sentence Selector framework (Lauriola and Moschitti, 2021), which is the state of the art for open-domain answer selection, we use a Transformer model to encode and rank triplets. The input of the transformer is encoded as *[CLS] t [SEP] $a_i$ [SEP] $q_i$ [EOS]*. Note that this allows to jointly modeling the semantic dependencies between the two questions, e.g., their similarity, and the rele-

vance of the answer to both questions.

**The DataBase** The DataBase employed in this work consists of questions and their answers collected from various heterogeneous public high-quality annotated open-domain QA datasets, including: GooAQ, WQA (Zhang et al., 2021), WikiAnswer (Fader et al., 2014), CovidQA (Möller et al., 2020), and HotpotQA (Yang et al., 2018). We extracted questions and the associated correct answer text span from these datasets, and ingested these q/a pairs into our DataBase.

Beyond annotated resources, we enhanced our database with various sets of artificially generated q/a pairs. First, we considered questions from QuoraQP. Differently from the other datasets above, QuoraQP is designed for question duplication detection task and not QA. Thus it simply consists of question pairs. Some answers are available for a small fraction of questions, which were selected through a heuristic approach based on the rank of users' content in Quora threads (Wang et al., 2020). To expand the QuoraQP, we collected the missing answers using a similar approach described by Zhang et al. (2021): given an input question, we queried a 2020 CommonCrawl snapshot[2] using BM25 and we selected the 200 most relevant documents. Then, documents are split into sentences and a state-of-the-art sentence selector (Lauriola and Moschitti, 2021) is used to select the top-ranked passage as the answer. In addition, the score returned by the passage reranker model can be effectively used to measure the likelihood of the answer being correct. We applied a threshold to this score and accepted only the top 10% of q/a pairs. This guarantees a higher level of answer quality in our database. We manually labeled 200 randomly selected answers from this unsupervised set as quality assessment, observing an accuracy of 93.0%. Finally, we also ingested q/a pairs from ELI5 (Fan et al., 2019). This dataset consists of questions collected from three subreddits for which the answers have been ranked by the users' up-votes in the thread. Although this heuristic removed part of the noise of the dataset, to ensure the maximum quality we keep only the 50% of the q/a pairs with the highest sentence selector score. After a manual annotation, we estimate the 84.3% of accuracy. Our final DB contains ≈ 6.3 millions of English questions and their correct answers pairs. Further details and statistics are reported in Table 1. It

| Source | QA | Q | Q length | A length |
|---|---|---|---|---|
| **Labeled** | | | | |
| GooAQ | 3.1M | 2.9M | $9.1_{\pm2.3}$ | $45.9_{\pm18.9}$ |
| WQA | 391K | 80.5K | $7.5_{\pm3.2}$ | $24.8_{\pm11.3}$ |
| WikiAnswer | 2.3M | 2.3M | $9.1_{\pm2.5}$ | $60.3_{\pm117.3}$ |
| CovidQA | 2K | 1.9K | $10.6_{\pm4.1}$ | $15.8_{\pm17.1}$ |
| HotpotQA | 64K | 64K | $20.4_{\pm10.6}$ | $4.1_{\pm2.4}$ |
| **Unlabeled** | | | | |
| Quora Match | 230K | 170K | $12.5_{\pm6.7}$ | $38.8_{\pm20.5}$ |
| QuoraQP | 219K | 134K | $9.6_{\pm2.9}$ | $25.1_{\pm10.8}$ |
| ELI5 | 58.9K | 58.7K | $17.6_{\pm9.10}$ | $60.7_{\pm28.1}$ |
| **Total** | **6.3M** | **5.7M** | | |

Table 1: Main statistics of QUADRo database, QA= q/a pairs, Q= unique questions.

is worth noticing that this DataBase can be combined with other Existing DataBases, including PAQ. However, we did not use those resources for two main reasons. First, PAQ authors stated that the resource contains a considerable amount of noise and errors (only 82% correctness). Second, PAQ consists of generated questions that are, sometimes, unnatural. In the next section, we describe how we used this DataBase to build our annotated resource. Thus, we prefer data quality and correctness over possible coverage/recall.

## 4 Annotated Dataset

As Section 2 highlights, existing resources are not suited to train models for DBQA. In this section, we describe the annotation workflow, emphasizing the main features and key aspects.

### 4.1 Annotation workflow

We randomly selected 15,211 questions from our DataBase, and removed them from it to avoid bias for the next experiments. For each of the questions above, we ran the QUADRo pipeline described in Section 3[3] and retrieved $k$ most similar questions and their answers. Based on the annotation budget and preliminary observations, we set $k = 30$ to balance high diversification (15,211 diverse input questions) and recall (see Figure 1). We used Amazon Mechanical Turk (AMT) to annotate retrieved triplets $(t, q_i, a_i)$, target question, retrieved question, and answer as correct/incorrect, where this binary label represents the semantic equivalence or not, respectively, between $t$ and $q_i$.

Given the complexity and partial ambiguity of the annotation task, we ensured the high quality

---

[2]https://commoncrawl.org/2020/?utm_sou

[3]We describe the configuration of the search engine used to collect q/a pairs in Section 5.

| Split | Q | QA | pos/neg QA |
|-------|------|--------------|---------------------|
| Train | 11711 | $28.9_{\pm 10.3}$ | $6.1_{\pm 7.9}$ / $22.7_{\pm 11.5}$ |
| Dev. | 1500 | $30_{\pm 0}$ | $4.8_{\pm 5.3}$ / $25.2_{\pm 5.3}$ |
| Test | 2000 | $30_{\pm 0}$ | $4.7_{\pm 5.1}$ / $25.3_{\pm 5.1}$ |

Table 2: Data splits. Q:#queries, QA:q/a pairs per query.

of our annotation by applying several procedures, some of which are completely novel. First, we used two distinct annotators for each triplet, and a third one to resolve the tie cases, thus the final label is computed with the majority vote. Second, we limited the annotation to turkers with documented historical annotation activities, including master turkers with at least 95% approval rate and 100 approved tasks. Third, we provide the following definition of equivalence to the annotators: two questions are equivalent iff they (i) have the same meaning AND (ii) share the same answers. While expressing the same meaning is theoretically enough to assess the semantic equivalence of the questions, in practice, there are several sources of ambiguity, e.g., it is well known that the meaning depends on the context, and the latter may be underspecified. The answer instead provides a strong context that can focus the interpretation of the annotator on the same meaning of both questions[4]. Finally, we introduced a set of positive and negative control triplets that we used to filter bad annotations.

Each annotation task consists of 7 triplets of which 2 control triplets, one negative and one positive. Positive control triplets are designed to be as clear as possible and sufficiently simple and negative triplets are automatically defined by selecting two random questions. Answering incorrectly to at least one of the control pairs caused the rejection of the annotation task. Moreover, if the same turker failed more than 10% of the total assigned HITs, all their HITs are discarded and the turker is blacklisted, precluding the execution of further annotations. Guidelines, examples, and further details are shown in the Appendix C. Some annotation anecdotes are shown in Appendix D.

### 4.2 Dataset analysis

One source of complexity was given by the fact that annotated triplets do not have 100% accurate answers as the internal accuracy of q/a pairs in the DataBase is $\approx 93\%$. Thus, we clarified with the annotators that they should use answers to help their judgment, but these are not necessarily cor-

---

[4]A few examples are reported in the appendix.

rect. We ran an independent annotation batch to evaluate the performance of the annotators and to quantify the benefits of including the answer into the annotation interface. Our manual evaluation of 200 annotations showed that adding the answer in the annotation workflow reduces the relative error rate for this task up to 45%, leading to an absolute accuracy of 94%. In other words, the answer is a precious source of information which, beyond the modeling part where previous work already showed the importance of the q/a relevance, can significantly help the human judgment and thus the quality of the dataset. To our knowledge, our is the first resource for question-question similarity exploiting the answer as context in order to perform accurate annotation.

Given all precautions taken to improve the quality of the data (MTurk filters, answer exposure, control triplets), we observed an agreement between the first two annotators on 78% triplets (random labeling corresponds to 50% agreement). We also measured the Cohen's kappa inter-annotator metric (Artstein and Poesio, 2008) to be 0.875, indicating good agreement quality. We do not have data showing the agreement between the 3 annotators as the third annotator was considered only when there was no agreement between the first two annotators. The resulting dataset consists of 15,211 unique input questions and 443K annotated triplets. The answerability, i.e. the number of input queries with at least one positive match in the set of associated q/a pairs, is 75.4%. On average, each question has 5.8 out of 30 positive pairs. We split our dataset into training, development, and test. Details of the split are shown in Table 2.

## 5 Experiments

We ran various distinct sets of experiments to assess the quality and challenges of our dataset and to show the empirical performance of state-of-the-art models in DBQA tasks. First, we analyze the retrieval and ranking tasks thanks to our new dataset. Then, we use this resource to fine-tune and evaluate retrieval and ranker models. Finally, we compare our approach against strong Web-based open-domain QA systems. As we previously mentioned, our experiments aim to empirically clarify and report some design choices of DBQA systems including, for instance, the modeling of the answer.

## 5.1 Model Training

We used our dataset to train and evaluate both, retrieval and ranking models.

**Retrieval Models:** we built the bi-encoder (described in Section 3) with RoBERTa continuously pre-trained on $\approx 180M$ semantic text similarity pairs[5]. We considered the following two input configurations: QQ: the model encodes the input question, $t$, and the evaluation questions $q_i$ in the first and second transformer branches (with shared weights). QQA: the model encodes $t$ in the first branch, and the concatenation of the $q_i$ and $a_i$ in the second one. After this procedure, the model computes the cosine similarity between the two generated representations. We first fine-tuned the models on QuoraQP with artificially generated answers as previously described. Then, we fine-tuned the resulting model on our ranking dataset.

**Reranking Models:** we start from the state-of-the-art sentence selector model proposed by Lauriola and Moschitti (2021). The model consists of an Electra-base trained on ASNQ on triplets of the form *[CLS] question [SEP] answer [SEP] context [EOS]*, where *context* refers to additional sentences that are relevant for the answer. The checkpoint was then fine-tuned on QuoraQP and lately on our dataset. As in our case the $a_i$ can be considered as the context of both $t$ or $q_i$, or alternatively, $q_i$ may be considered the context of $t$ and $a_i$. Therefore, the use of a pre-trained model with context is promising. Similarly to the retrieval model training, we considered various configurations, including (i) QQ only using $(t, q_i)$ pairs; (ii) QAQ corresponding to $(t, a_i, q_i)$, where the question acts as a context for the answer, (iii) QQA corresponding to $(t, q_i, a_i)$, where $a_i$ is the context, and (iv) QA encoding $(t, a_i)$, i.e., a standard sentence selector for QA. For both, retrieval and reranker the model score ranks the q/a pairs with respect to each input question.

## 5.2 DBQA performance

As discussed in Section 4.1, we used an initial retrieval model to select the q/a pairs that compose the question reranking dataset. Based on preliminary experiments conducted on existing benchmarks, we used a Sentence-RoBERTa (base) model. The model is trained on QuoraQP with QQA configuration. Similarly, we trained an initial reranker

---

[5]See appendix A for further pre-training details.

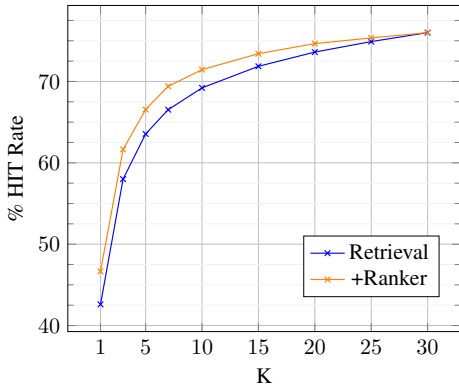

Figure 1: Hit rate of the retrieval module and the end-to-end system (QUADRo).

with QAQ configuration to be tested on our collected dataset. Figure 1 shows the Hit-rates at $k$, defined as the portion of questions with at least one positive in the top $k$ returned pairs, of the sole retrieval component and the Hit-rate of the entire system, i.e., after the reranking performed by the answer selector, applied to the 30 retrieved (and annotated) answers.

We note that: First, as we used the retrieval mentioned above to select 30 question/answer pairs to be annotated in our dataset the plot exactly indicates the retrieval performance. Since the ranker operates on just the 30 annotated candidates, we can also evaluate its impact on an end-to-end system in an open-domain QA setting. This means that the accuracy of our DBQA system (Hit@1) in answering open domain questions sampled (and removed) from the DB is 46.6%, as indicated in the picture. Second, as expected, the larger the $k$, the higher is the probability of retrieving a good answer. Using 30 candidates, the retrieval is able to find an answer for 75.4% of the questions. The reranker boosts the retrieval performance by 4% absolute (42.6 vs 46.6). Then, it consistently improves the retrieval. Note that a configuration entirely based on the answer selector (i.e. without retrieval) is infeasible due to the complexity of the model as it needs to run a Transformer for each stored pair (6.3M). Finally, there is a considerable gap between the system accuracy (i.e.: Hit@1) and the potential accuracy (Hit@30). This suggests that our dataset enables challenging future research. Also, there is a significant amount of unanswerable queries, which opens other research challenges on how learning to not answer to improve system F1.

## 5.3 Retrieval and reranker evaluation

We fine-tuned our models on the collected dataset as described in 5.1 and evaluated them on the test split. Details of the fine-tuning step are described in Appendix B. Table 3 reports the performance of (i) *S-RoBERTa$_{QQA}$: the initial neural retrieval model used to select annotation data (and thus not trained on this resource), (ii) S-RoBERTa$_x$: our retrieval models fine-tuned on the collected resource, and (iii) Electra$_x$: the fine-tuned reranking models. As mentioned before, these models can be applied to the top 30 candidates retrieved by *S-RoBERTa$_{QQA}$, which are all annotated thus enabling their easy evaluation. The selection of RoBERTa for retrieval and Electra for reranking was driven by a set of preliminary experiments described in Appendix B.

We note that the Electra (reranker) generally outperforms S-RoBERTa (retrieval), as it is a cross-encoder, where a single network encodes the whole triplet. Differently, S-RoBERTa uses two different encoders which provide two uncontextualized representations that are successively combined only in the final layer. Moreover, S-Roberta using the QQA configuration highly outperforms QQ (+5.0 P@1), while this gap disappears for Electra since its cross-encoder seems to be enough powerful to compensate for the lack of context, i.e., the answer.

Concerning Electra models, QQA is outperformed by QQ (-0.7 P@1), mostly because the checkpoint that we used was trained by Lauriola and Moschitti (2021) on QA tasks, thus the model expects the answer to be close to the query. Indeed, QAQ, which is closer to how the checkpoint was trained, improves the P@1 of QQ by 0.8. It should also be stressed the fact that our dataset has been annotated with respect to question-question equivalence and not question-answer correctness. Although the answers were shown to the annotators, they were biased on the question and used the answer just as support. Finally, we evaluated existing state-of-the-art question de-duplication models (Reimers and Gurevych, 2019) consisting on a RoBERTa cross encoder trained on QuoraQP[6] (see table 3). Not surprisingly, our models based on similar architectures achieve better performance thanks to the fine-tuning on the target data. We did not evaluate other existing solutions (e.g. Wang et al. (2020)) as models are not publicly available.

---

[6]Public checkpoints are available https://www.sbert.net/docs/pretrained_cross-encoders.html.

| Model | P@1 | MAP | MRR |
|---|---|---|---|
| *S-RoBERTa$_{QQA}$ | 39.1 | 39.1 | 50.4 |
| S-RoBERTa$_{QQ}$ | 43.4 | 41.6 | 52.9 |
| S-RoBERTa$_{QQA}$ | $48.4_{\pm0.4}$ | $45.6_{\pm0.4}$ | $58.3_{\pm0.4}$ |
| Electra$_{QA}$ | $37.1_{\pm0.6}$ | $40.4_{\pm0.2}$ | $49.5_{\pm0.3}$ |
| Electra$_{QQ}$ | $50.0_{\pm0.2}$ | $47.7_{\pm0.3}$ | $59.5_{\pm0.2}$ |
| Electra$_{QQA}$ | $49.3_{\pm0.2}$ | $47.63_{\pm0.1}$ | $59.2_{\pm0.1}$ |
| Electra$_{QAQ}$ | $\mathbf{50.8_{\pm0.2}}$ | $\mathbf{48.4_{\pm0.1}}$ | $\mathbf{60.2_{\pm0.1}}$ |
| QP-RoBERTa$_{base}$ | 43.5 | 41.8 | 54.4 |
| QP-RoBERTa$_{large}$ | 45.6 | 43.5 | 56.0 |

Table 3: Experiment results on the proposed dataset. (*) This model is the one used to build the dataset. QP-models are state-of-the-art cross encoders (Reimers and Gurevych, 2019).

## 5.4 Comparison with Web-based QA and Large Language Models

In the previous experiments, we (i) tuned retrieval and ranking models on task-specific annotated data and we (ii) empirically showed what the best configurations are to improve accuracy, corroborating the findings presented in the existing literature and aligning with the state-of-the-art in DQD and DBQA. As a further step, we evaluated the end-to-end DBQA pipeline updated with models tuned on our dataset, and we compared it against (i) a popular open-book Web-based QA system (WebQA), and (ii) Large Language Models (LLM).

WebQA consists of a search engine, BING, which finds a set of documents relevant to the input question, and a state-of-the-art sentence selector, which chooses the most probably correct answer sentence, extracted from the retrieved documents. This is a typical web-based QA pipeline, the main difference with existing work is that we used BING, a complete and complex commercial search engine, instead of standard solutions based on BM25 or DPR (Zhang et al., 2022a,b). For LLMs, we used two popular models, Falcon 7B (ZXhang et al., 2023) and Vicuna 7B v1.5 (Chiang et al., 2023) to which we asked to generate a proper answer for the given query exploiting their parametric knowledge.

Our DBQA configuration consists in S-RoBERTa-QQA used as the retriever and Electra-base-QAQ as the reranker. For both, DBQA and WebQA, the answer selectors are applied on top of the top K candidate sentences from the search engines. After preliminary experiments, we set K = 500. The retrieval corpus for QUADRo is the DataBase of 6.3M q/a pairs we described before, while, the WebQA corpus is defined as the retrieval capability of BING.

LLM baselines, i.e. Falcon and Vicuna, were used in zero-shot setting. The models tried to generate an answer given an input question through their parametric knowledge, without accessing extra grounding information. The networks received a prompt to generate a natural and correct answer given the input request, with a few examples. The complete prompt is shown in the Appendix H.

To remove biases, we evaluated them with three new sets of questions from open domain sources: Quora, Natural Questions (Kwiatkowski et al., 2019), and TriviaQA (Joshi et al., 2017), of size 200, 200, and 150, respectively. See Appendix F for further details regarding these datasets. The questions selected for this test are not in our DataBase nor in the training set. We manually annotated the correctness of the returned answers.

The results are reported in Table 4. We note that: First, QUADRo highly outperforms WebQA on Quora 58% vs 35%. The main reason is that finding relevant documents for open domain questions is always challenging, especially for questions such as Quora where the context is typically underspecified. In contrast, QUADRo can find questions that have similar shape and meaning, and whose answer is still valid for the target question. In the same way, Quadro also outperforms Vicuna (+33%), while having the same performance on Falcon (58%). The main reason is that both LLMs rely only on their parametric knowledge, which can lead to hallucinations and incorrect answers.

Second, the rationale above is confirmed by the better performance of WebQA on NQ. In this case, the questions that we used have always a relevant Wikipedia page by the construction of the NQ dataset. BING is enough powerful to find these pages so that the only real challenge is the answer selection step. QUADRo still produces a good accuracy (50%), while Falcon and Vicuna are still not able to retrieve the correct information from their knowledge resulting in a poor accuracy of 40%. Finally, QUADRo also outperforms both Vicuna and WebQA on TriviaQA questions. The latter are rather challenging questions, which are difficult to find on the web and complex to be answered and understand by LLM without grounding. QUADRo seems to be able to generalize its DB content enough well when operating the search step, or at least better than what a standard search engine can do over web documents. Furthermore, it's worth highlighting that Falcon 7B seems to be strong enough to prop-

| Model | Quora | NQ | TriviaQA |
|---|---|---|---|
| WebQA | 35.0 | **56.0** | 27.0 |
| Falcon 7B | **58.0** | 40.0 | 40.6 |
| Vicuna 7B v1.5 | 25.0 | 40.0 | 21.0 |
| QUADRo | **58.0** | 50.5 | **29.3** |
| - our dataset | 53.0 | 47.0 | 28.0 |
| - neural SE | 39.0 | 37.5 | **29.3** |
| - sentence sel. | 51.5 | 40.0 | 19.0 |
| - answer relev. | 50.0 | 45.0 | 25.3 |

Table 4: End-to-end accuracy of QUADRo and other baselines in 3 open domain QA settings. The best results are highlighted in bold characters. "- x" means QUADRo without the feature "x".

erly understand the complexity of the questions generating appropriate answers.

## 5.5 Ablated end-to-end comparisons

For completeness, we ran an ablation study to evaluate the impact of each component of our system. The last four rows of Table 4 show the end-to-end open-domain QA accuracy of QUADRo: (i) without fine-tuning on our dataset, emphasizing the importance of the resource; (ii) when substituting the neural search engine with a standard BM25; (iii) without the answer selector (neural search engine only); and (iv) with neural SE and selector trained on standard question/question similarity, without using the answer. Similarly to the previous comparison, the sentence selectors were applied on top 500 retrieved q/a pairs.

The results show that each element is extremely relevant, and the ensembling of various key technologies makes QUADRo an effective DBQA system. For example, training on our dataset increases the accuracy by 5% on Quora, 3.5% on NQ, and 1.3% on TriviaQA. The latter small improvement is probably due to the fact that the questions of TiviaQA are rather specific, thus training on general data would impact less than on other datasets.

The usage of the neural search engine is essential, our QUADRo using BM25 performs 19% absolute less than when using our neural ranker. This is rather intuitive as the major advantage of using embeddings is their ability to generalize short text, such as the one constituting questions, while TF×IDF heuristics based on lexical features of BM25 largely suffer from sparse representations.

The answer selector component provides as expected a good contribution, 7-10% absolute for all

datasets. Finally, the usage of the answer representation as context is rather impactful, from 4% to 8%. This demonstrates that, when the target of the evaluation is answer correctness instead of question equivalence, models that take answer context into account are clearly superior.

## 6 Conclusions

End-to-end QA pipelines that find the answers in a precompiled DataBase of question/answer pairs (DBQA) have become popular in recent years due to their efficiency, accuracy, and other benefits. One of the main issues of these approaches is the lack of large training/test data to develop competitive models. Existing resources are limited in dimension, scope, or they do not consider the relevance of the answer. In order to fill this gap, we introduce a novel annotated resource to train models for DBQA. Our dataset comprises 443,000 examples, making it one of the largest annotated resources for this task. Another novelty is the annotation mechanism that considers both, the similarity of the question with respect to the input and the relevance of the answer. Furthermore, our experiments report various key aspects of DBQA systems often neglected in the literature, including the contribution of the answer in the retrieval and ranking tasks and the need for high-quality training data. We believe that our data and models, made available to the research community, will enable interesting future research in optimizing models for DBQA. For example, how to improve the selection of correct q/a pairs in the top $k$ pairs.

## 7 Limitations

The most glaring limit of QUADRo is that the possibility of answering a question strictly depends on the coverage of the database of question/answer pairs. Although the database can be enlarged, e.g. by incorporating q/a pairs from PAQ, covering even more questions, there is still a conceptual limit that prevents the system from answering very infrequent questions. Moreover, the database requires mechanisms for content refresh as some questions might change the answer over time.

Concerning the retrieval model, a largely known issue of Dense Retrieval systems regards possible drop in performance subject to data domain shift (Wang et al., 2021). Although (i) we train the models on various pre-training tasks and open-domain questions, and (ii) our end-to-end experiments show competitive performance with new data, we cannot quantify this possible issue.

Finally, we observed that, despite the possibility of reading an answer, annotators tend to focus more on the query-question similarity and less on the query-answer relevance. A possible consequence is that models trained on triplets instead of query-question pairs may experience a degradation in performance due to skewed labels. Notwithstanding this observation, models trained on query-question pairs work poorly in end-to-end QA evaluation (see Table 4).

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

## A  Pre-training

Starting from a public checkpoint of our search engine based on Sentence-RoBERTa-base, we continuously pre-trained it on a plethora of datasets for unsupervised STS tasks (paraphrasing, sentence similarity, question answering, and summarization...). These datasets include MS-MARCO (Nguyen et al., 2016), Natural Questions, The Semantic Scholar Open Research Corpus (Lo et al., 2020), PAQ (Lewis et al., 2021), NLI (Bowman et al., 2015), Altex (Hidey and McKeown, 2016), AmazonQA (Gupta et al., 2019), CNN Dailymail (See et al., 2017), Coco Captions (Lin et al., 2014), CodeSearchNet (Husain et al., 2019), Eli5 (Fan et al., 2019), Fever (Thorne et al., 2018), Flickr30K (Young et al., 2014), GooAQ, Sentence Compression (Filippova and Altun, 2013), SimpleWiki (Coster and Kauchak, 2011), Specter (Cohan et al., 2020), SQuaD (Rajpurkar et al., 2016), StackExchange (Narayan et al., 2018), WikiHow (Koupaee and Wang, 2018), and Xsum (Narayan et al., 2018).

These datasets consist of pairs of semantically equivalent texts (e.g.: question and answer, title and abstract of a document, paraphrasing...). Overall, the pre-training data includes $\approx 180$M positive (i.e. semantically related) text pairs and $\approx 17.5$M existing hard-negatives[7]. We consider a simple pre-training task where the model predicts if two texts are semantically equivalent or not. We used the MultipleNegativeRanking (Henderson et al., 2017) loss on top of the bi-encoder model combined with cosine similarity as the distance metric in order to make the model able to learn powerful embeddings for retrieval. We used a batch size of 384 and a max sequence length of 256 tokens. We use the AdamW optimizer with a learning rate of $2e^{-5}$.

## B  Reproducibility details

We used the same strategy to fine-tune all models in the experiments shown in this paper. We used the development split of our proposed resource to select the optimal hyper-parameters configuration through a grid search and to early stop the training when observing a degradation of the validation loss for 2 consecutive epochs.

The hyper-parameters evaluated for the retrieval (Sentence-RoBERTa) are: the learning rate $\{5e-6, 1e-5, 2e-5, 5e-5\}$, the batch-size $\{64, 128, 256, 384\}$, and the max sequence length $\{128, 256\}$. Concerning ranking models (Electra), we tuned the learning-rate $\{5e-6, 1e-5, 3e-5, 5e-5\}$ and the batch-size $\{64, 128, 1024\}$ the max sequence length is set to 256 tokens.

The selection of the initial checkpoint and architecture for the retrieval (RoBERTa) and the reranking (Electra) components was driven by a set of preliminary experiments, where we evaluated multiple Transformer models. We observed, and reported on

---

[7]Hard negatives are provided for a small subset of pre-training datasets

| Model | P@1 | MAP | MRR |
|---|---|---|---|
| RoBERTa$_{QAQ}$ | $47.5_{\pm 0.3}$ | $46.5_{\pm 0.2}$ | $57.7_{\pm 0.1}$ |
| Electra$_{QAQ}$ | $\mathbf{50.8_{\pm 0.2}}$ | $\mathbf{48.4_{\pm 0.1}}$ | $\mathbf{60.2_{\pm 0.1}}$ |

Table 5: Electra and RoBERTa results comparison on the question/answer reranking dataset. The suffix $QAQ$ indicates the input setting.

| Model | Pearson corr. | Spearman corr. |
|---|---|---|
| S-RoBERTa | 85.4 | 85.1 |
| S-Electra | 74.8 | 75.1 |
| S-Deberta-V3 | 76.4 | 77.3 |

Table 6: S-RoBERTa vs S-Electra and S-Deberta-V3 performances comparison on the STSbenchmark dataset. All models are base architectures.

table 5, that Electra performs better than RoBERTa (+3.3 P@1, +1.9 MAP, and +2.5 MRR.) as cross-encoder in the ranking task measured on our test set. We consider the QAQ configuration as is it the one with the better performance. Both models have the same pre-training as described in the previous sections.

Differently, Sentence-RoBERTa showed better retrieval performance compared to Sentence-Deberta-V3 (He et al., 2021) and Sentence-Electra on measured on STSbenchmark (Cer et al., 2017), an estabilished benchmark for these tasks.

## C    Annotators training and guidelines

Annotators were asked if two input questions were equivalent or not. The definition of equivalence is: that two questions are equivalent iff they have the same intent/meaning and share the same answers. A possible answer for the second question was provided to help the judgment.

We provided a set of guidelines with detailed and explained examples to train the annotators. Guidelines consist of a clear description of the task and a set of positive and negative query-question-answer triplets. The examples are meant to clarify when query-question pairs can be considered duplicated and how the answer can be used to help the judgment. Some examples used to train the annotators are reported in Table 7.

Alongside the guidelines, we introduced a set of control triplets in order to guarantee high annotation quality. Control triplets are designed to be clear, simple, and easy to be judged as positive or negative. Table 8 reports a subset of control triplets used during the annotation.

Annotators were rewarded with 0.15$ per annotation task.

## D    Anecdotes

This section shows and discusses some examples of produced annotations, shown in Table 9. In the first example, Case A, the query and question are closer in terms of question-wording and are asking for the same thing making it easy to label them as duplicates even without taking the answer into account. By contrast, in Case B we can notice that query and question seem to be asking for different things. In this case, the answer played a key role for the final annotation label, since it answers both the query and the question. The assigned positive label is then correct.

Case C contains an annotation error. In this case, the query is contained in the question. While the question is asking for the name of the back and the front of a boat, the query is asking only for the name of the back of a boat. Without considering the answer, an annotator might be prone to label these questions as non-duplicated, since they are asking for slightly different concepts. However, in an end-to-end QA setting, the answer is correct for the input query (and the question), and thus we should consider the triplet as positive.

In the last example, Case D, we report a wrong positive annotation. The query and the questions seem to be identical, but they are not. The query is asking how to replace the battery from a liftmaster remote control, while the question is asking how to replace the battery of a liftmaster remote keypad. In this case, the query and the question are referring to two distinct devices. Moreover, the provided answer is not correct with respect to the query. In this case, the correct annotation should be negative.

## E    Latency

We conducted a latency analysis on QUADRo, evaluating the efficiency with respect to various key aspects, including the number of retrieved q/a pairs and the size of the database. In our tests, we used (i) a Nvidia A100 GPU, (ii) a S-RoBERTa (base) retrieval model with an embedding size of 768, and (iii) an Electra-base reranker.

Figure 2 show the latency of an end-to-end request when varying the number of the retrieved q/a pairs while using a database of $\approx$ 6.3M q/a pairs. As you can see, the time scales linearly with the amount of retrieved data. To retrieve and rerank 500 q/a pairs QUADRo took only $\approx$ 0.53s, 140ms to retrieve and rerank 50 pairs. According to Figure 2 we can notice that the majority of the time is

| Positive Examples | Negative Examples |
|---|---|
| **Query**: Can a cat and a dog get along? 
 **Question**: Do cats like the company of dogs and in the other way around? 
 **Answer**: If you are lucky, your cat and dog can become friends within a couple of hours. But that won't usually happen. It takes time for cats to adapt to the dogs and similarly for the dogs to learn how to behave around cats. 
 **Explanation**: *These questions are both asking if Cats and dogs can be friends. The Answer for the Question is also correct for the Query* | **Query**: Who did kill Brutus? 
 **Question**: Who did Brutus kill? 
 **Answer**: Brutus was one of the leaders of the conspiracy that assassinated Julius Caesar 
 **Explanation**: *These questions are not asking for the same thing. Moreover, the Answer for the Question is not correct for the Query* |
| **Query**: Can a person fall in love with another person while he/she is already in love? 
 **Question**: Is it possible for people to love 2 person at the same time? 
 **Answer**: It is possible to love and be intimate with more than one person at a time. 
 **Explanation**: *These questions are both asking if loving 2 people at the same time is possible. The Answer for the Question is correct for both Question and Query.* | **Query**: What is the best restaurant in LA ? 
 **Question**: What is the best dish of the best restaurant in LA? 
 **Answer**:The best dish of the best restaurant in LA is Lobster Rolls 
 **Explanation**: *Those questions are not asking for the same thing. The query asks for a restaurant while the Question asks for a dish. Moreover, the Answer is not correct for the Query* |

Table 7: Explained examples used during annotators training.

| |
|---|
| **Query:** What is the color of the sun? 
 **Question:** Which color the sun has? 
 **Answer:** The sun has a temperature of 5800 Kelvin, so it appears white 
 **Label:** positive |
| **Query:** What is the food of Koalas? 
 **Question:** What do Koalas eat? 
 **Answer:** Eucalyptus 
 **Label:** positive |
| **Query:** What is the coldest place in the world? 
 **Question:** What shape is a watermelon? 
 **Answer:** The watermelons are round or oval shaped 
 **Label:** negative |
| **Query:** How many humans are there in the world? 
 **Question:** What is the color of the strawberry? 
 **Answer:** Typically they are red 
 **Label:** negative |

Table 8: Examples of control triplets used to discard annotations.

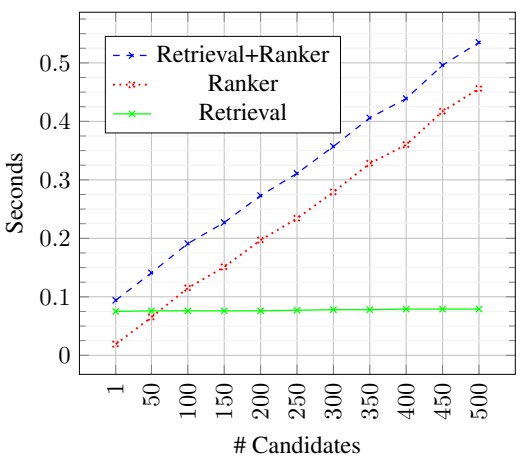

Figure 2: Latency of the end-to-end QUADRo system while increasing the number of retrieved and reranked q/a pairs. Values are averaged over 200 executions.

spent in the ranking process rather than in retrieving the pairs from the database. The retrieval time is ≈ 77ms, and it does not scale as the number of returned q/a pairs increases.

Figure 3 shows the latency of the system while increasing the dimension of the DB. In the experiment, we set the number of retrieved q/a pairs to 500. The retrieval time scales as the size of the database increases. Is worth mentioning that the retrieval process is in the magnitude order of milliseconds, confirming the efficiency of our system.

We release the implementation and the code to replicate these experiments upon request. The experiments were based on PyTorch 1.9, Cuda 10.2, Python 3.8, and Transformers (by HuggingFace) 4.10.

## F Datasets

We ran most of our experiments on our collected dataset for question ranking. We divided our dataset into training, development, and test questions as described in Table 2.

Additionally, we ran end-to-end QA evaluations on questions sampled from various open-domain sources, including QuoraQP and:

**Natural Questions**: NQ (Kwiatkowski et al., 2019) is a popular open-domain QA dataset which consists of questions sampled from Google traffic. A Wikipedia page containing a long and a short answer is associated with each question. Consequently, each question is virtually answerable by a web-based QA system.

| Case # | Examples |
|--------|----------|
| Case A | **Query**: how old are oldest fossils of organisms? 
 **Question**: The oldest fossils date to how long ago? 
 **Answer**: The oldest documented fossil organisms date to roughly 3.8 billion years ago, shortly after the period of heavy asteroid bombardment in Earth's history. 
 **Label**: 1 |
| Case B | **Query**: where did the term bully pulpit come from 
 **Question**: Who or what is meant by the expression Bully Pulpit? 
 **Answer**: This phrase, used by Theodore Roosevelt, refers to the office of presidency. A pulpit is a place to preach from and "bully" is an older adjective meaning excellent. Therefore a bully pulpit is a great place to make speeches from . Roosevelt appreciated the fact that people listened to him when he spoke as the President. 
 **Label**: 1 |
| Case C | **Query**: What is the name called of a back end of boat? 
 **Question**: What is the name for the back and front of a boat? 
 **Answer**: The front of a vessel be it a ship or boat, is the bow or stem. The back is the stern. 
 **Label**: 0 |
| Case D | **Query**: how how to replace battery in liftmaster remote control? 
 **Question**: how to change battery in liftmaster remote keypad? 
 **Answer**: 'Slide the battery cover down. The battery is located at the bottom of your keypad.', 'Disconnect the old battery and remove it. Remember to dispose of your old battery correctly.', 'Install the new battery. ... ', 'Put the battery cover back in place.' 
 **Label**: 1 |

Table 9: Annotated triplets from our dataset.

**TriviaQA**: TriviaQA (Joshi et al., 2017) is a QA dataset containing over 95,000 open-domain q/a pairs authored by trivia enthusiasts and independently gathered evidence documents. Trivia questions are designed to be more challenging, complex, and compositional compared to the other datasets.

## G Metrics

We measure the performance of QA systems with Accuracy in providing correct answers, i.e., the percentage of correct responses, which also refers to Precision-at-1 (P@1) in the context of reranking. We also use standard metrics for ranking: Mean Average Precision (MAP), Mean Reciprocal Rank (MRR), and Hit-rate@k, which measure the percentage of questions that have at least one correct answer (or correct question) in the top-k retrieved/ranker items.

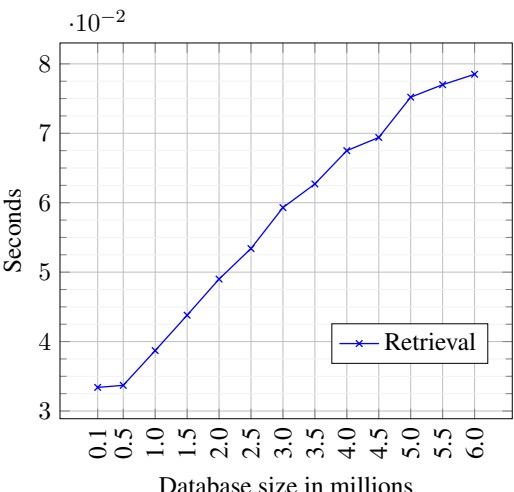

Figure 3: Latency of the end-to-end QUADRo system while increasing the dimension of the DB. Values are averaged over 200 executions.

## H LLM zero-shot prompt for Q&A

We used the following prompt while generating answers for our experiments through Falcon and Vicuna.

*"Answer the question below. The answer must be [well-formed and concise]. The only accepted format is the following:*
*Question: [here the question]*
*Answer: [here the answer]*
*Here you have some examples:*
*Example 1:*
*Question: what is an apple?*
*Answer: An apple, (Malus domestica), is a domesticated tree and fruit of the rose family (Rosaceae), one of the most widely cultivated tree fruits. Apples are predominantly grown for sale as fresh fruit, though apples are also used commercially for vinegar, juice, jelly, applesauce, and apple butter and are canned as pie stock.*
*Example 2:*
*Question: What is the largest airport in the world by travelers?*
*Answer: Atlanta Hartsfield-Jackson International Airport (ATL) is the larger airport in the world with 75,704,760 total passengers. Dubai International Airport (DXB) is the second busiest airport, followed by Tokyo International Airport (HND) which is the third.*
*Ok, let's begin!*
*Question: {input-question}"*