# OpenReview forum: "QUADRo: Dataset and Models for QUestion-Answer Database Retrieval"
_EMNLP/2023/Conference — EMNLP 2023 Findings_

### Official Review · Reviewer_XhtM · 2023-08-02

**Soundness:** 4

**Excitement:**

4: Strong: This paper deepens the understanding of some phenomenon or lowers the barriers to an existing research direction.

**Paper Topic And Main Contributions:**

This paper focuses on the task of question-answer database retrieval (DBQA), which given an input question consists in retrieving the answer from a database of question-answer pairs. The paper clearly introduces the task and describes its challenges, and accordingly, presents as main contribution a new large annotated dataset to train and evaluate models for DBQA. In addition, it reports an extensive experimentation to assess the quality and characteristics of the dataset, as well as to show its potential to be used in open-book QA and to fine-tune retrieval and ranker models.

**Questions For The Authors:**

- In Section 3, in the part of the “search engine”, wouldn't it make more sense to take only pairs that exceed a minimum established similarity? what values of k were tested?

- In Section 4.1, in how many cases was the third annotator required Although some percentages are mentioned, I would recommend reporting standard measures of inter-annotator agreement.

- In the same Section 4.1, several strategies were followed to ensure the quality of the annotation, but how much did they really contribute to improving its final quality? Was there any evaluation of this?

- In 4.2, the analysis was done on a sample of 200 instances, how was this size determined?




**Reasons To Accept:**

Being a paper whose contribution is a new resource:

- It clearly exposes the limitations of current resources and the needs of the task that are intended to be met with this new dataset.
- The methodology used to build the recourse is clear and seems appropriate. In this regard, the new dataset is larger than previous similar ones, but above all it exploits the answer as context in order to improve the quality of the annotations.
- Several experiments are presented to support its quality and relevance, as well as experiments that support its use in other QA tasks and subtasks.

In general, my main reason for supporting its acceptance is that it can contribute to improving and extending DBQA research.




**Reasons To Reject:**

Without being an expert on the subject of the paper, I can't find any compelling reason to reject the work. However, I find two negative aspects:

- The sections describing the experiments are not as clear as the first three sections, for example, several of the tables and figures were not explained in sufficient detail.
- The paper makes strong criticisms about the quality of the previous datasets, but does not present strong evidence (quantitatively speaking) of the quality of the new resource. Several of the claims in this regard are made based on a manual analysis of only 200 instances (which is not a statistically significant sample).



**Reproducibility:**

2: Would be hard pressed to reproduce the results. The contribution depends on data that are simply not available outside the author's institution or consortium; not enough details are provided.

**Reviewer Confidence:**

2: Willing to defend my evaluation, but it is fairly likely that I missed some details, didn't understand some central points, or can't be sure about the novelty of the work.

---

> ### Author Rebuttal · Authors · 2023-08-29
>
> We would like to thank the reviewer for their comments and suggestions.
>
> > The sections describing the experiments are not as clear as the first three sections, for example, several of the tables and figures were not explained in sufficient detail.
>
> Thanks for the feedback. We  plan to invest the extra page for the camera ready version to improve the clarity of the experiments section and the captioning of figures and tables.
> In addition, we will add more examples to emphasize the quality of the dataset.
>
> Below we answer your questions:
>
> > In Section 3, in the part of the “search engine”, wouldn't it make more sense to take only pairs that exceed a minimum established similarity? what values of k were tested?
>
> The retrieval model is less powerful and so less accurate than the ranker component. This means it can be less precise in identifying relevant q/a pairs. For this reason, a similarity threshold may filter out some best candidates that only the reranker can identify (as the top candidates). Additionally, using a threshold, in corner cases, can lead to have no candidates for an input query, which in some scenario may not be acceptable, e.g., when suggesting similar questions to users.
>
>
>  > in Section 4.1, in how many cases was the third annotator required Although some percentages are mentioned, I would recommend reporting standard measures of inter-annotator agreement.
>
> It was needed 22% of cases. In the camera ready, we will also add standard inter-annotator measures. Thanks for the great suggestion.
>
> > In the same Section 4.1, several strategies were followed to ensure the quality of the annotation, but how much did they really contribute to improving its final quality? Was there any evaluation of this?
>
> We used multiple strategies to ensure high quality. Some of these were based on standard and established approaches, including adoption of master turkers (annotators with established high quality work history) with >95% acceptance rate on previous annotation tasks. Studying the quality of an annotated resource without these constraints can be expensive and it is outside the scope of the paper. Regarding other employed strategies, we estimated that control questions were able to filter 20-21% of annotation tasks. We will add this data to the paper to improve its completeness.
>
>  > In 4.2, the analysis was done on a sample of 200 instances, how was this size determined?
>
> We used a sample of 200 instances per annotation strategy (qq vs qqa) for 400 annotations in total. This number is sufficient as the result of a t-test showed that the distribution of errors (using our methods or not) are indeed different with high confidence, p<0.1. Moreover, evaluating the correctness of annotations is a time-consuming procedure that requires a significant amount of effort (annotation, fact verification, grammar check, analysis and understanding of the results, e.g., different outcome).
>
> Regarding reproducibility, we believe it is a key aspect of a resource paper such as ours.
> We would like to highlight that we will release our source code, dataset and database to the scientific community. We will add a GitHub link in the paper.

---

### Official Review · Reviewer_pEq3 · 2023-08-03

**Soundness:** 3

**Excitement:**

4: Strong: This paper deepens the understanding of some phenomenon or lowers the barriers to an existing research direction.

**Paper Topic And Main Contributions:**

The paper introduces a new resource named QUADRo which is a dataset for evaluating QA systems which retrieves question-answer pairs given queries (DBQA). Compared to previous work the author incorporate the answers into the consideration of question-question similarity, combine several topical QA sets and resulting generated dataset is fast greater than previous works (443K).
The authors claim that this resource is the largest that is suitable for training database QA systems, that was also humanly annotated.

An additional contribution included in the paper is a pipeline built using the dataset to evaluate the contribution of annotation, the addition of fine-tuning of the models on the dataset and the contribution of including the answers for retrieval/ranking.

**Reasons To Accept:**

* dataset comprises of multiple sources with various question and answer length, and combinations of artificially generated question-answer pairs
* authors made sure to reach high percentage of accuracy for generated samples (by human annotation).
* annotation workflow well defined and executed to verify high quality human annotation.
* empirical results showing that the dataset improves retrieval and ranking  compared to baselines, including to some degree on new domains (WebQA) and comparing to a commercial search engine (BING)

**Reasons To Reject:**

* despite using multiple data sources in QUADRo, most of the samples probably came from GooAQ and WikiAnswer (5.4M question-answer pairs out of 6.3M), making it not very heterogeneous.
* to narrow choice of baseline models for analysis of a resource paper

**Reproducibility:**

3: Could reproduce the results with some difficulty. The settings of parameters are underspecified or subjectively determined; the training/evaluation data are not widely available.

**Reviewer Confidence:**

4: Quite sure. I tried to check the important points carefully. It's unlikely, though conceivable, that I missed something that should affect my ratings.

**Typos Grammar Style And Presentation Improvements:**

L56: grammar
DataBase - casing? consider -> database or index
Table 1: lenght ->length

---

> ### Author Rebuttal · Authors · 2023-08-29
>
> We would like to thank the reviewer for their thoughtful comments
>
> > despite using multiple data sources in QUADRo, most of the samples probably came from GooAQ and WikiAnswer (5.4M question-answer pairs out of 6.3M), making it not very heterogeneous.
>
> We believe this is a good point: although the questions are from open domains and thus they are rather general by construction, GooAQ and WikiAnswer may have bias that are inherited by our DB. GooAQ and WikiAnswer represent the predominant part of the DB, but we also use other datasets that mitigate the bias and improve the heterogeneity, namely WQA, ELI5, QuoraQP, Quora Match, WikiAnswer, CovidQA and HotpotQA.
> In the future, we will further improve the heterogeneity of our database adding more datasets from other resources (e.g., from CCQA dataset, about 130M q/a pairs).
>
>  > to narrow choice of baseline models for analysis of a resource paper
>
> Secondly, we will add other baselines models in the camera ready of the paper (if accepted), including LLMs, e.g., (Vicuna 7B and Falcon 7B).

---

### Official Review · Reviewer_VGzs · 2023-08-05

**Soundness:** 4

**Excitement:**

3: Ambivalent: It has merits (e.g., it reports state-of-the-art results, the idea is nice), but there are key weaknesses (e.g., it describes incremental work), and it can significantly benefit from another round of revision. However, I won't object to accepting it if my co-reviewers champion it.

**Paper Topic And Main Contributions:**

The authors built a large annotated dataset for DBQA, i.e. finding answers from a precompiled question-answer database for a given question. They believe that answers are important for retrieval and ranking in DBQA, and therefore, consider not only question-question similarity but also question-answer similarity and correctness in the dataset construction and the proposed pipeline QUADRo. Extensive experiments verified the positive contribution of answers in candidate retrieval and answer ranking for DBQA and the need of high-quality training data in building an effective DBQA system.

**Questions For The Authors:**

1. Why and how do you select 15,211 questions from the big collection of 5.7M questions? What about their distribution over different sources?

**Reasons To Accept:**

1. A large annotated general dataset is created for Question Answering based on pre-compiled question-answer databases. It is expected to be a good resource for DBQA R&D in the future.

2. Answers are proposed to be included in the annotated dataset and utilized in the QUADRo pipeline for DBQA. Extensive experiments demonstrate its effectiveness for candidate retrieval and answer ranking.

**Reasons To Reject:**

1. The key structure of the proposed retrieval-reranking DBQA architecture QUADRo is inspired by previous work (Lewis et al. 2021; Seonwoo et al. 2022), which is stated in line 233.

2. There are  not a few typos or grammar errors. Please refer to the "Typos Grammar Style And Presentation Improvements" section below.

**Reproducibility:**

4: Could mostly reproduce the results, but there may be some variation because of sample variance or minor variations in their interpretation of the protocol or method.

**Reviewer Confidence:**

4: Quite sure. I tried to check the important points carefully. It's unlikely, though conceivable, that I missed something that should affect my ratings.

**Typos Grammar Style And Presentation Improvements:**

Typos or grammar errors:

1. in abstract, "An effective approach to design ...", "either do not consider answers or their quality in the annotation process.", "train end evaluate"
2. line 031, "from on large corpus ..."
3. line 059, "without re-train models"
4. line 240, "QUEstion"
5. line 304, "using as a similar approach described by"
6. line 335, "authos"
7. line 384, "the same meaning both questions"
8. line 420, "our is ..."
9. line 430, "between the first annotators"
10. lines 560 and 635, "enough powerful"
11. line 677, "7-10% absolute ..."

---

> ### Author Rebuttal · Authors · 2023-08-29
>
> We would like to thank the reviewer for their professional review.
>
> >The key structure of the proposed retrieval-reranking DBQA architecture QUADRo is inspired by previous work (Lewis et al. 2021; Seonwoo et al. 2022), which is stated in line 233.
>
> Although our retrieval-ranking DBQA architecture takes inspiration from previous work by Lewis et al. 2021 and Seonwoo et al. 2022, there are some key differences:
>
> First, those architectures were based on query-question similarity only, while our ranking and retrieval components use also the answer (of the retrieved question) to improve the similarity between query and question.
> Specifically, we use the triplet, (Query, Question, Answer), which, especially in the cross-encoder transformer (used for the reranking), is a completely new model. Indeed, the neural cross-attention is able to combine information matched between the query and question with the query and answer and question and answer. This way, our model learn to rank questions also by matching common patterns between (query, answer) and (question, answer), i.e., using textual relations between pairs.
>
> Second, both the models have been trained on our datasets, which was designed for question-question retrieval. This is different than datasets for standard question-question matching, since the model can learn how to rank questions from our dataset rather than how to classify pairs of questions to output if they are a good match or not.  We plan to extend the retrieval-ranking architecture in our future work.
>
> In addition, we would like to clarify that the main contribution of our work is the resources that we release to train models for question/answer retrieval and ranking. Dataset, Database, and source code will be released with the paper to make the replication of our experiments easier and support the research in this field.
>
> > Why and how do you select 15,211 questions from the big collection of 5.7M questions?
> > What about their distribution over different sources?
>
> We randomly sampled 15211 questions from two distinct sources: ~30% of the total from the QuoraQP dataset and ~70% from the WQA dataset. The datasets are widely used for question deduplication and for open domain question answering tasks, respectively. They are considered foundational research datasets.
>
> Many thanks to indicate typos and grammar errors: we will fix them for the camera ready

---

### Meta-Review · Area_Chair_LqYs · 2023-09-17

**Recommendation:** 4

**Metareview:**

This paper creates a large human-annotated dataset and models for "question-answer database retrieval", where given a question, retrieval takes place of the relevant answers from a pre-existing database with question-answer pairs.

Pros:
- Introduces a valuable, solidly annotated data set that will be helpful for research in question-answer retrieval.
- The data set doesn't only allow for comparing new questions with a set of existing questions, but also for looking at the answers to those pre-existing questions in order to determine whether a given pre-existing (question+answer) pair could be relevant to the new question.
- Dataset and newly trained models are released.

Cons:
- Presentation could stand to be improved somewhat; details around the experiments aren't as clear as the other sections.
- Some inter-annotator measures are missing.
As per the review thread, authors would be addressing these in the camera-ready.

---

### Decision · Program_Chairs · 2023-10-07

**Decision:**

Accept-Findings

**Comment:**

This paper creates a large human-annotated dataset and models for "question-answer database retrieval", where given a question, retrieval takes place of the relevant answers from a pre-existing database with question-answer pairs.

Pros:
- Introduces a valuable, solidly annotated data set that will be helpful for research in question-answer retrieval.
- The data set doesn't only allow for comparing new questions with a set of existing questions, but also for looking at the answers to those pre-existing questions in order to determine whether a given pre-existing (question+answer) pair could be relevant to the new question.
- Dataset and newly trained models are released.

Cons:
- Presentation could stand to be improved somewhat; details around the experiments aren't as clear as the other sections.
- Some inter-annotator measures are missing.
As per the review thread, authors would be addressing these in the camera-ready.